# Longer Context, Deeper Thinking: Uncovering the Role of Long-Context Ability in Reasoning

**Wang Yang[1], Zirui Liu[2], Hongye Jin[3], Qingyu Yin**
**Vipin Chaudhary[1], Xiaotian Han[1]**
[1]Case Western Reserve University [2] University of Minnesota - Twin Cities
[3]Texas A&M University
{wxy320,vipin,xhan}@case.edu, zrliu@umn.edu, jhy0410@tamu.edu

## Abstract

Recent language models exhibit strong reasoning capabilities, yet the influence of long-context capacity on reasoning remains underexplored. In this work, we hypothesize that current limitations in reasoning stem, in part, from insufficient long-context capacity, motivated by empirical observations such as *i)* higher context window length often leads to stronger reasoning performance, and *ii)* failed reasoning cases resemble failed long-context cases. To test this hypothesis, we examine whether enhancing a model's long-context ability before Supervised Fine-Tuning (SFT) leads to improved reasoning performance. Specifically, we compared models with identical architectures and fine-tuning data but varying levels of long-context capacity. Our results reveal a consistent trend: models with stronger long-context capacity achieve significantly higher accuracy on reasoning benchmarks after SFT. Notably, these gains persist even on tasks with short input lengths, indicating that long-context training offers generalizable benefits for reasoning performance. These findings suggest that long-context modeling is not just essential for processing lengthy inputs, but also serves as a critical foundation for reasoning. We advocate for treating long-context capacity as a first-class objective in the design of future language models. Our code is anonymously available at https://github.com/uservan/LCTMerge.

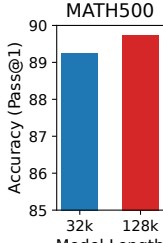 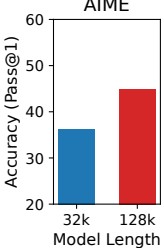 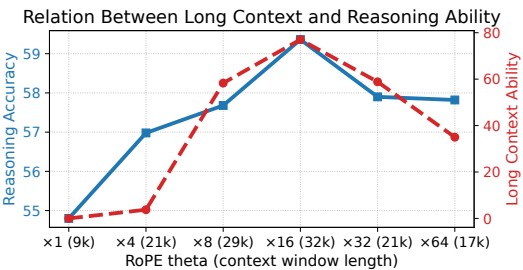

Figure 1: Impact of long-context capacity on mathematical reasoning. **Left:** Accuracy (Pass@1) on MATH500 and AIME datasets for public models with 32k and 128k context lengths, showing consistent improvements in reasoning performance with longer context windows. The 32k and 128k LLMs refer to three different public models, as shown in Table 1. **Right:** Reasoning accuracy versus RoPE theta values, highlighting a strong correlation between long-context capacity and reasoning performance. Increasing the RoPE theta value typically extends the effective context window length.

39th Conference on Neural Information Processing Systems (NeurIPS 2025).

# 1 Introduction

Large language models (LLMs) have recently demonstrated impressive reasoning capabilities across a wide range of benchmarks [1–3]. Despite this progress, the underlying factors that influence such reasoning abilities remain only partially understood. One particularly underexplored dimension is long-context ability—the model's capacity to utilize a longer reasoning path during inference—which could affect the reasoning performances and the reasoning model fine-tuning.

While prior work has primarily focused on training paradigms and dataset quality to enhance reasoning [4–6], we hypothesize that a model's reasoning ability is also fundamentally constrained by its long-context capacity. This hypothesis is grounded in three empirical observations. *First*, models with extended context lengths (e.g., 128k vs. 32k) consistently achieve higher accuracy on reasoning benchmarks such as MATH500 and AIME, suggesting a direct performance benefit from stronger long-context modeling (Table 1, Figure 1). *Second*, case studies reveal that failed generations often involve extremely long outputs, sometimes truncated at generation limits, and exhibit issues like repetition or incorrect cross-referencing—failure patterns strongly linked to inadequate long-context capability (Figure 2, Figure 3). *Third*, modern reasoning datasets now include many samples exceeding 8K or even 10K tokens—substantially longer than early CoT data—requiring models to learn from long, variable-length reasoning sequences (Figure 4). Together, these findings underscore long-context capacity as a critical factor for reasoning ability, and motivate the central question: *Does improving a model's long-context ability during pretraining enhance downstream reasoning?*

To rigorously investigate this, we conduct a controlled study comparing language models with identical architectures and fine-tuning data, but varying degrees of long-context pretraining. Our experimental results reveal a consistent and compelling trend: models with stronger long-context capabilities consistently outperform their counterparts on reasoning tasks after SFT. Notably, these improvements extend to reasoning problems with short input lengths, suggesting that long-context training imparts generalizable cognitive benefits that go beyond simply processing long sequences. As shown in Figure 1, `LLaMA3-8B-Instruct` exhibits varying reasoning performance after training when equipped with different levels of long-context capability. Notably, reasoning ability tends to increase or decrease in accordance with the strength of the model's long-context capacity, suggesting a direct correlation between the model's reasoning ability and long-context capacity.

Based on our experimental results, we thus propose a *Recipe for Reasoning Fine-Tuning*, which advocates for *appropriately enhancing a model's long-context capacity prior to reasoning SFT*—for instance, by extending its context length to 128K tokens. Applying this recipe to `Qwen2.5-Math-7B-Instruct`, we observe substantial improvements: performance on `MATH500` increases from an average of $85.04$ to $88.70$, and on `AIME`, from $15.00$ to $28.00$.

# 2 Motivation: behavioral evidence suggests a connection between long context and reasoning

In this section, we present a set of empirical observations that suggest a strong behavioral connection between a model's long-context capability and its reasoning performance. Through controlled comparisons, token length analyses, and failure case studies, we observe that LLMs with stronger long-context abilities not only perform better on reasoning benchmarks but also handle diverse and extended reasoning sequences more reliably. These findings collectively highlight long-context modeling as a key factor and component in enabling strong reasoning ability.

## 2.1 Higher context window length often leads to stronger reasoning performance.

Recent advances in long-context modeling have enabled language models (LLMs) to process substantially longer sequences. However, it remains unclear whether such long-context capacities yield tangible benefits for reasoning tasks. In this section, we empirically examine the relationship between long-context ability and reasoning performance. We collect a set of well-known open-source reasoning models fine-tuned from `Qwen/Qwen2.5-7B-Instruct`. These models are categorized into two groups based on their long-context capacity: 32k and 128k tokens. We then evaluate and compare their reasoning performance on two math reasoning benchmarks: `MATH500` and `AIME`.

The detailed results are reported in Table 1. Figure 1 presents that models with longer context lengths (128k vs. 32k) consistently achieve higher accuracy on mathematical reasoning benchmarks such as MATH500 and AIME. This suggests that the ability to encode and maintain longer contextual dependencies can directly translate into better reasoning capabilities. These results collectively highlight the importance of effective long-context training—not only for tasks involving long inputs, but also for general reasoning even when test-time inputs are relatively short.

Table 1: Performance comparison on MATH500 and AIME benchmarks for some popular open-source reasoning models with different long-context abilities at 32k and 128k context lengths. Reasoning models with enhanced long-context capacity (128k) generally exhibit improved performance, particularly on the AIME benchmark. Averages are reported in the bottom row.

| Long Context Ability at 32k | MATH500 | AIME | Long Context Ability at 128k | MATH500 | AIME |
|---|---|---|---|---|---|
| OpenR1-Qwen-7B | 90.36 | 43.11 | DeepSeek-R1-Distill-Qwen-7B | 91.68 | 45.56 |
| OpenThinker-7B | 86.80 | 25.78 | OpenMath-Nemotron-7B | 94.00 | 74.67 |
| OpenThinker2-7B | 90.60 | 40.22 | DeepMath-Zero-7B | 83.48 | 14.67 |
| OpenThinker3-7B | 93.72 | 64.00 | AceReason-Nemotron-7B | 93.84 | 62.89 |
| Avg (32k) | 90.37 | 43.28 | Avg (128k) | 90.75 | 49.45 |

**Questions:** Alice chooses a set A of positive integers. Then Bob lists all finite nonempty sets B of positive integers with the property that the maximum element of B belongs to A. Bob's list has 2024 sets. Find the sum of the elements of A.

**Failed Solution:** Alright, so I have this problem here: Alice chooses a set A of... which has 8 ones.
k=10, 8≠10.
k=11:2035,9 ones.
k=12:2036,8 ones.
k=13:2037,9 ones. ...

**Questions:** Three spheres with radii 11, 13, and 19 are mutually externally tangent. A plane intersects the spheres in three congruent circles centered at A, B, and C, respectively, and the centers of the spheres all lie on the same side of this plane. Suppose that $AB^2 = 56$. Find $AC^2$.

**Failed Solution:** Okay, let me try to work through this problem step by step. So...Let's let's let's denote:
Let's let's let's let's let's let's let's let's let's let's let's let's let's let's let's let's let's let's let's let's let's let's let's let's let's let's let's let's let's let's ...

Figure 2: **Case Study: Repetition Failure.** Two failure cases where the model produces clearly repetitive sentences in its answers. Such repetition is a common symptom of insufficient long-context capability, leading to strange responses and degraded reasoning quality in extended sequences.

**Questions:** The expression $2 \cdot 3 \cdot 4 \cdot 5 + 1$ is equal to 121, since multiplication is carried out before addition. However, we can obtain values other than 121 for this expression if we are allowed to change it by inserting parentheses. For example, we can obtain 144 by writing $(2 \cdot (3 \cdot 4)) \cdot (5 + 1) = 144$. In total, how many values can be obtained from the expression $2 \cdot 3 \cdot 4 \cdot 5 + 1$ by inserting parentheses?

**Failed Solution:** Okay, so I have this math problem here. It's about ...
1. All multiplications first, then addition: 121.
2. Group 2 * (3 * (4 * (5 + 1))) + 1: 127.
3. Group (2 * (3 * 4)) * (5 + 1): 144.
4. Group 2 * ((3 * (4 * 5)) + 1): 42.
5. Group 2 * ((3 * 4) * (5 + 1)): 144....

**Questions:** Let $f(x)$ be an odd function, and let $g(x)$ be an even function. $f(f(g(f(g(f(x))))))$ is even, odd, or neither?

**Failed Solution:** Alright, so I've got this problem here: ...
2. $g(f(x)) = (-x)^2 = x^2$: even.
3. $f(g(f(x))) = f(x^2) = -x^2$: which is odd.
4. $g(f(g(f(x)))) = g(-x^2) = (-x^2)^2 = x^4$ : even.
5. $f(g(f(g(f(x))))) = f(x^4) = -x^4$ : which is odd
6. $g(f(g(f(g(f(x)))))) = g(-x^4) = \ldots$

Figure 3: **Case Study: Contextual Reference Failures.** Two failure cases where the model makes incorrect references to expressions introduced earlier in the problem. These errors occur in the later stages of the response and reflect a typical symptom of insufficient long-context capability.

## 2.2 Case study: failed reasoning cases resemble failed long-context cases.

We analyze failed cases of reasoning models and long-context inputs, revealing a strong connection between reasoning ability and long-context capacity. Specifically, we examine the average input lengths of correct and incorrect predictions made by three reasoning models—DeepSeek-R1-Distill-Qwen-1.5B, 7B, and 14B—on two math benchmarks: MATH500 and AIME. As shown in Figure 4, incorrect predictions are typically associated with inputs exceeding 10k tokens, suggesting that these failures may stem from insufficient long-context handling.

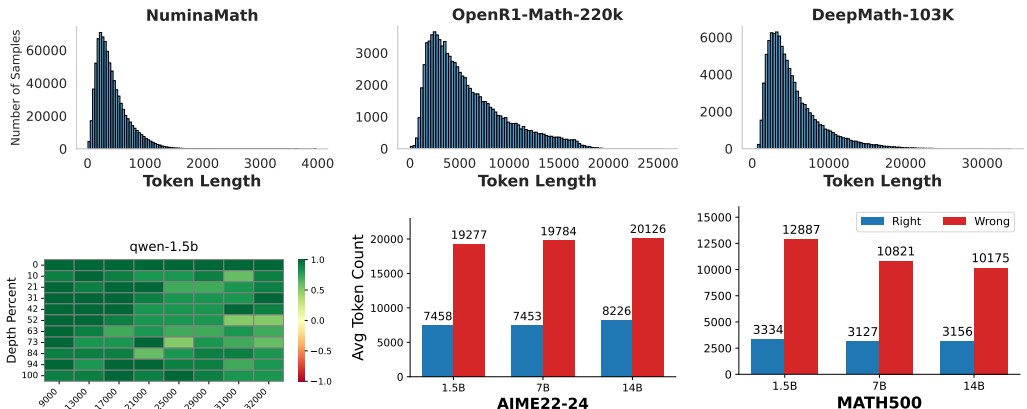

Figure 4: **Top:** Length distribution of three reasoning datasets. `NuminaMath-CoT` represents early chain-of-thought (CoT) data with short sequences, while `OpenR1-Math-220K` and `DeepMath-103K`, generated by `DeepSeek-R1`, exhibit significantly longer outputs. **Bottom-left:** Performance of `DeepSeek-Distilled-Qwen-1.5B` on the *Needle-in-a-Haystack* benchmark with 32K context. **Bottom-middle/right:** Average output lengths of correct and incorrect generations on `AIME` and `MATH500` for `DeepSeek-Distilled-Qwen-1.5B`, 7B, and 14B. Incorrect answers consistently exhibit longer output lengths, indicating potential limitations of long-context ability in reasoning.

To further investigate this hypothesis, we manually inspect a subset of long-output failures and identify two recurring patterns, repetition and contextual reference failures, due to limited long-context capability. The first pattern involves excessive repetition, where the model loops over the same sentence or phrase, failing to advance the solution, as shown in Figure 2. The second pattern arises in the latter part of the output, where the model incorrectly recalls earlier mathematical expressions from the problem, leading to flawed reasoning and incorrect conclusions, as shown in Figure 3. Both failure modes underscore the model's struggle to maintain coherence and accuracy over long sequences—a well-known limitation of inadequate long-context capacity that degrades and distorts in-context learning performance in LLMs.

## 2.3 Long and variable reasoning data necessitate long-context models.

Current reasoning models are typically fine-tuned on long chain-of-thought (CoT) datasets generated by large-scale reasoning models. A key characteristic of these models is their tendency to produce *long and variable-length outputs*. As a result, the resulting datasets exhibit broad length distributions, with many examples exceeding 10K tokens—far longer than early CoT-style data. This necessitates that fine-tuned models be capable of handling such long and diverse sequences. In Figure 4, we analyze the length distributions of three representative datasets: `NuminaMath-CoT`, `OpenR1-Math-220K`, and `DeepMath-103K`. `NuminaMath-CoT`, representing early CoT-style data, primarily contains samples under 1K tokens. In contrast, `OpenR1-Math-220K` and `DeepMath-103K`, both collected from `DeepSeek-R1`, contain significantly longer reasoning sequences, with a large proportion of samples exceeding 4K tokens and some even surpassing 10K tokens, which is totally different with early CoT datasets like `NuminaMath-CoT`.

There is a lack of research exploring how such increased sequence lengths interact with or depend on the model's long-context capability after the release of Deepseek-R1. For instance, if a model lacks sufficient long-context ability, training on long reasoning sequences may fail to yield expected improvements—or even negatively impact performance—due to the model's inability to effectively utilize the full input. While modern models are advertised to handle long sequences (e.g., `Qwen2.5-1.5B-Instruct` supports up to 32K tokens), their effective context length is often substantially shorter. As shown in Figure 4, we evaluate `Qwen2.5-1.5B-Instruct` on the *Needle-in-a-Haystack* benchmark and observe that the model fails to maintain high accuracy across all cases in 32k contexts, indicating its limitations in effective long-context processing and long-context ability.

Table 2: Effective context length and long-context benchmark performance of `LLaMA3-8B-Instruct` under different `RoPE theta` scaling factors. We report the estimated effective context length, Needle-in-a-Haystack (NIAH) retrieval accuracy at 32k, as well as performance on LongBench and RULER. Results show that scaling up to RoPE ×16 consistently improves long-context robustness across all benchmarks, but further scaling (e.g., ×32 and ×64) leads to diminishing or even negative returns. Notably, the effective length surpasses the maximum sequence length (16k) of the current training dataset when the factor exceeds 4, which is sufficient for reasoning training.

| RoPE theta | ×1 | ×4 | ×8 | ×16 | ×32 | ×64 |
|---|---|---|---|---|---|---|
| Effective Context Length | 9k | 21k | 29k | 32k | 21k | 17k |
| 32k NIAH Score | 0.00 | 3.75 | 58.30 | 77.05 | 58.86 | 35.00 |
| LongBench Score | 21.14 | 39.21 | 39.78 | 40.41 | 38.96 | 38.01 |
| RULER Score | 56.13 | 69.57 | 79.62 | 94.24 | 88.07 | 84.98 |

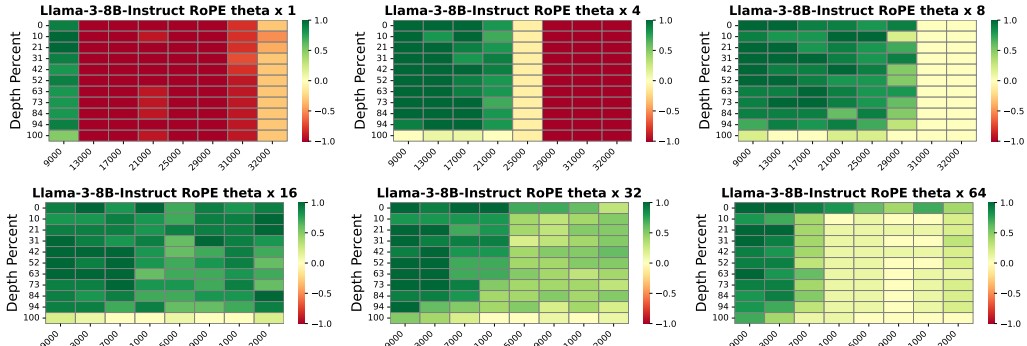

Figure 5: Needle-in-a-Haystack Results for LLaMA-3-8B-Instruct. Performance of LLaMA-3-8B-Instruct on the *Needle-in-a-Haystack* benchmark with a 32K context under different `RoPE theta` scaling factors. `RoPE theta x 16` refers to scaling the original `RoPE theta` by a factor of 16.

## 3 Empirical analysis: verifying the connection between reasoning and long-Context ability

In this section, we investigate the connection between a model's long-context capability and its reasoning performance. We begin by applying long-context extension strategies to obtain models with varying levels of long-context ability at 32k tokens. These models are then fine-tuned on different reasoning datasets using supervised fine-tuning (SFT), allowing us to assess how different context capacities influence the effectiveness of reasoning training. Next, we further extend the long-context capability to 128K tokens or beyond, in order to examine whether extreme context lengths can provide additional gains in reasoning performance. Finally, we introduce a recipe for LLM reasoning training: extend the long-context ability before reasoning finetuning and conduct an experiment to verify that this recipe is useful and effective. The overall training pipeline is illustrated in Figure 6.

### 3.1 Experimental setup

**Long context ability extension strategy.** We use two strategies to enhance long-context capability. The first is directly scaling the `RoPE theta` parameter by different factors, which has been shown to improve a model's ability to handle longer sequences [7]. The second leverages model merging: we merge the target model with another model that possesses stronger long-context capabilities. In this setting, we carefully control the merge ratio to ensure that the base performance remains nearly unchanged, allowing us to isolate the effect of long-context enhancement as the only influential factor.

**Data processing for reasoning SFT**. We utilize the OpenR1-Math-220K dataset [8] and divide it into two categories based on response length: **short** samples (responses within 8K tokens) and **long** samples (responses ranging from 8K to 16K tokens). For both categories, we sample 20K instances and perform correctness filtering to ensure that each response is factually accurate and correct. These two subsets are then used independently to fine-tune models to improve their reasoning ability.

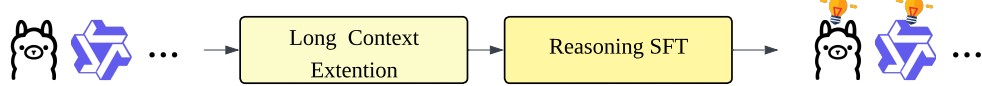

Figure 6: Pipeline for verifying the connection between reasoning and long-context ability. We first expand the model's long-context capability to obtain variants with different levels of long-context capacity. Then, we perform SFT on reasoning datasets to obtain reasoning-enhanced models.

**Training details**. All models are fine-tuned using four NVIDIA H200 GPUs. We employ the `LLaMAFactory` library with a batch size of 32, a learning rate of $1.0 \times 10^{-5}$ and 3 epochs.

**Long context evaluation**. We adopt the *Needle in a Haystack* benchmark provided by the OpenCompass framework to access long context ability. For simplicity and robustness, we use the accuracy on the single-haystack setting as our primary metric: a correct response receives a score of $+1$, a repetitive/degenerate answer receives $-1$, and an incorrect but non-degenerate response receives $0$.

**Reasoning evaluation**. To further evaluate the model's reasoning ability post-training, we use three math benchmarks: MATH500, AIME22–24, and GSM8K. Following the evaluation methodology from DeepSeek-R1, we adopt the *pass@1(5)* metric, where five responses are generated for each question and accuracy is computed over all the responses. This provides a more stable estimate of reasoning performance and abilities of Large Language Models after finetuning with datasets.

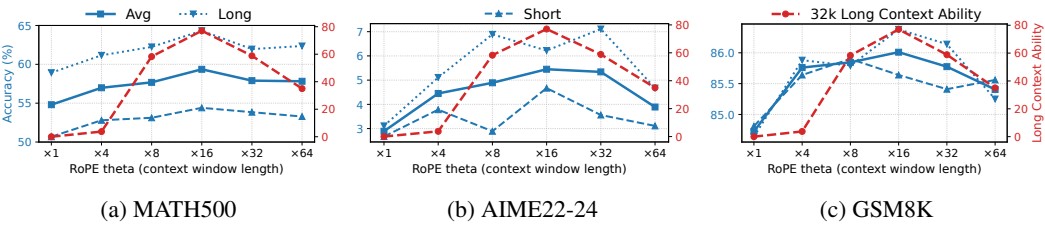

|     (a) MATH500     |     (b) AIME22-24     |     (c) GSM8K     |

Figure 7: Visualization of the relationship between 32k long-context ability and reasoning performance across different model variants of `LLaMA-3.1-8B-Instruct` on three math benchmarks (MATH500, AIME22-24, GSM8K). `Short` and `Long` refer to performance after fine-tuning on short and long reasoning datasets. `Avg` represents the average of the short and long fine-tuned results.

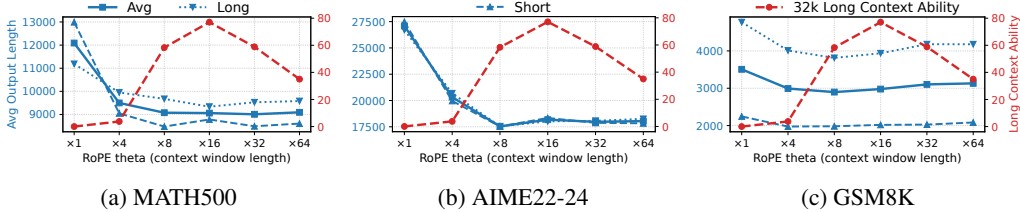

|     (a) MATH500     |     (b) AIME22-24     |     (c) GSM8K     |

Figure 8: Visualization of the relationship between 32k long-context ability and average output length across different model variants of `LLaMA-3.1-8B-Instruct` on MATH500, AIME22-24, GSM8K.

### 3.2 How does long-context capacity affect reasoning SFT?

In this experiment, we investigate how different levels of long-context capability affect the performance of reasoning SFT (Supervised Finetuing). We use `LLaMA3-8B-Instruct` as the base model, which originally supports up to 8K tokens. To obtain models with varying long-context capabilities, we scale the `RoPE theta` value by the following different factors: $\{1, 4, 8, 16, 32, 64\}$.

First, we evaluate the long-context ability of each variant using the *Needle-in-a-Haystack* benchmark. As shown in Figure 5 and Table 2, performance improves as the `RoPE theta` scaling factor increases from 1 to 16, reaching peak performance at 16. Beyond this point, further increases in `theta` lead to performance degradation. What's more, when the scaling theta is more than 4, the model's effective length is more than 16k, which is enough for the reasoning training for current datasets.

Table 3: **Effect of RoPE theta scaling on long-context and reasoning performance on MATH500.**
Evaluation of `LLaMA3-8B-Instruct` with different `RoPE theta` scaling factors on the 32K *Needle-in-a-Haystack* benchmark. `Base` refers to the model's performance before SFT, while `Short` and `Long` denote results after fine-tuning on short and long reasoning datasets, respectively. `Avg` represents the average of the `Short` and `Long` performances.

| RoPE | 32k long | | Acc(%) | | | | Average Output Length | | |
|---|---|---|---|---|---|---|---|---|---|
| theta | ctx ability | Base | Short | Long | Avg | Base | Short | Long | Avg |
| ×1 | 0 | **24.40** | 50.68 | 58.92 | 54.80 | 746 | 12991 | 11187 | 12089 |
| ×4 | 3.75 | 20.40 | 52.80 | 61.16 | 56.98 | 754 | 9047 | 9949 | 9498 |
| ×8 | 58.30 | 18.96 | 53.12 | 62.24 | 57.68 | 767 | **8476** | 9673 | 9075 |
| ×16 | **77.05** | 17.28 | **54.40** | **64.32** | **59.36** | 608 | 8782 | **9335** | 9059 |
| ×32 | 58.86 | 15.24 | 53.84 | 61.96 | 57.90 | 490 | 8489 | 9526 | **9007** |
| ×64 | 35.00 | 14.20 | 53.28 | 62.36 | 57.82 | 453 | 8605 | 9579 | 9092 |

Table 4: Cross-domain evaluation of long-context ability on GPQA (science) and Livecode (code).

| Task | Setting | GPQA Accuracy (%) | | | Livecode Accuracy (%) | | |
|---|---|---|---|---|---|---|---|
| | | ×1 | ×4 | ×16 | ×1 | ×4 | ×16 |
| 32k NIAH | - | 0.00 | 3.75 | 77.05 | 0.00 | 3.75 | 77.05 |
| Accuracy | before training | 31.82 | 31.31 | 30.81 | 12.30 | 10.16 | 7.45 |
| | after training | 37.27 | 39.19 | 41.92 | 25.34 | 27.95 | 32.27 |

Next, we perform supervised fine-tuning (SFT) using both `short` and `long` reasoning datasets on models with different `RoPE theta` settings. The detailed results are summarized in Table 3. In addition, we plot the relationship between each model's long-context capability and its reasoning performance. We also visualize how long-context ability correlates with the average output length. The results are shown in Figures 7 and 8. We observe that the best reasoning performance across all three evaluation benchmarks consistently occurs when the `RoPE theta` scaling factor is set to 16—coinciding with the strongest long-context capability observed in the Needle-in-a-Haystack task. In contrast, models with suboptimal `theta` settings exhibit reduced accuracy undergoing the same SFT procedure. These findings suggest that enhancing long-context processing ability can directly contribute to improved reasoning performance after reasoning SFT (Supervised Fine-tuning).

Furthermore, we observe that, in general, models fine-tuned on `long` reasoning datasets outperform those fine-tuned on `short` datasets in both `MATH500` and `AIME`. These long-form datasets typically contain more complex problems and richer intermediate reasoning steps, making them more effective for improving the model's reasoning ability. However, in order to benefit from such data, models must possess sufficiently strong and effective long-context capabilities to process the extended inputs.

### 3.3 Generality and robustness of long-context gains to reasoning ability

We investigate whether the benefits of stronger long-context ability generalize across tasks, model families, and input lengths. Our results show that the improvements are not limited to math reasoning.

**Cross-domain generalization.** We fine-tuned `LLaMA3-8B-Instruct` models with RoPE scales ×1, ×4, and ×16 on 20k samples from the *science* and *code* domains of `OpenThoughts3-1.2M`. Evaluations on GPQA and Livecode (Table 4) show that stronger long-context ability consistently improves accuracy beyond math reasoning.

**Across model families.** We trained `Phi-4` (14B) with the same RoPE settings. Consistent gains on MATH500 (Table 5) confirm that the benefit is not tied to a specific architecture or scale. The results of AIME22-24 are in Table 10 and it has the similar performance like MATH500.

**Short-input reasoning.** On the short-input `MMLU-STEM` benchmark, models with stronger long-context ability also achieve higher post-training accuracy (Table 6). This indicates that long-context training not only preserves but can reinforce reasoning on short-input tasks.

Table 5: Performance of `Phi-4` under different RoPE scales on MATH500.

| Model | RoPE | 32k NIAH (%) | MATH500 (before, %) | MATH500 (after, %) |
|---|---|---|---|---|
| Phi-4 | ×1 | 52.27 | 79.52 | 88.62 |
| Phi-4 | ×4 | 78.07 | 77.48 | 89.14 |
| Phi-4 | ×16 | 84.77 | 73.20 | 89.90 |

Table 6: Effect of long-context ability on short-input reasoning (MMLU-STEM) with `LLaMA3-8B`.

| RoPE Scale | 32k NIAH (%) | MMLU-STEM (before, %) | MMLU-STEM (after, %) |
|---|---|---|---|
| ×1 | 0.00 | 54.36 | 71.06 |
| ×4 | 3.75 | 53.85 | 73.04 |
| ×16 | 77.05 | 51.44 | 74.27 |

Overall, these results demonstrate that the gains from stronger long-context ability are robust: they extend beyond math to science and code, hold across LLaMA, Qwen, and Phi model families, and even benefit short-input reasoning.

## 3.4 Does extremely long context bring further gains on reasoning SFT?

In previous experiments, we extended models to handle up to 32K tokens and observed notable improvements in reasoning performance. A natural question arises: *can even stronger long-context capabilities further enhance reasoning*, or is there a limit beyond which performance saturates or even degrades? To explore this, we conduct experiments using models with a context length of 1M tokens (Qwen2.5-7B-Instruct-1M)—far beyond the typical range of existing reasoning datasets.

We adopt a linear merging strategy to construct models with varying long-context capacities while minimizing changes to their base ability. Specifically, we merge two models with different context capabilities at various ratios to obtain models with intermediate long-context strengths. With selected merge ratio, we ensure that the base capabilities remain largely unchanged, isolating long-context capacity as the key variable. We apply this strategy to two models: Qwen2.5-7B-Instruct-1M with long context ability at 1M and Qwen2.5-7B-Instruct with long context ability at 32k.

We first evaluate the merged models on *Needle-in-a-Haystack* at 32k and find that long-context ability grows with the merge ratio, but near-perfect scores make it less discriminative (Appendix A). We therefore adopt more challenging 32k tasks such as *Value Tracking* and *Question Answering*, which better capture effective long-context processing by requiring stronger long-range reasoning.

We fine-tune the merged models on both `short` and `long` reasoning datasets, and evaluate on `MATH500`, `AIME22-24`, and `GSM8K`. We also examine the relationship between effective long-context ability, reasoning accuracy, and output length (Figures 9 and 10). Results show that moderate merge ratios (e.g., 0.1, 0.7) yield strong effective long-context ability and high reasoning accuracy, while the 1M model (ratio 1.0) exhibits weaker effective long-context utilization and degraded performance.

## 3.5 Proposed reasoning recipe: extend context length first

Based on our previous experiments, we propose a training **recipe** for improving reasoning capabilities via supervised fine-tuning (SFT): *first appropriately enhance the model's long-context capability, then apply reasoning-specific SFT*. This recipe aims to better prepare the model for long-form reasoning tasks, where both context length and reasoning complexity are critical. To validate this approach, we experiment with the `Qwen2.5-Math-7B-Instruct` model, which demonstrates strong mathematical performances but has a limited context length of 4k tokens, which is a good example of the recipe.

We enhance its long-context capability by multiplying the `RoPE theta` value by a scaling factor of 16 and merging the model with `Qwen2.5-7B-Instruct-1M` using a merge ratio of 0.3. After the merging step, the long-context ability improves noticeably, as is shown in Appendix A. Subsequently, we perform SFT using both `short` and `long` reasoning datasets, and evaluate the fine-tuned models on two benchmarks: `MATH500` and `AIME22-24`. As shown in Table 7, across both short and long

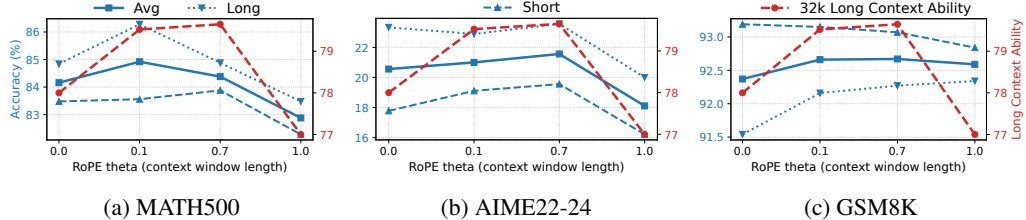

Figure 9: Visualization of the relationship between 32k long-context ability and reasoning performance across different model variants of `Qwen2.5-7B-Instruct` on MATH500, AIME22-24, GSM8K. A merge ratio of 0.1 indicates that the long-context variant (`Qwen2.5-7B-Instruct-1M`) contributes 10% to the final merged model. `Short` and `Long` refer to performance after fine-tuning on short and long reasoning datasets. `Avg` represents the average of the short and long fine-tuned results.

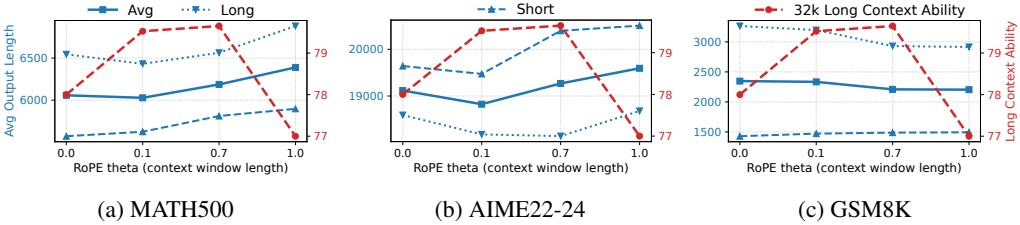

Figure 10: The relationship between 32k long-context ability and average output length across different model variants of `Qwen2.5-7B-Instruct` on three math benchmarks.

reasoning datasets, the model variant with `RoPE theta` scaled by 16 and a merge ratio of $0.3$ consistently achieves the highest performance after reasoning SFT on short and long datasets.

## 4 Related Works

**Reasoning models.** The release of `DeepSeek-R1` [9] has catalyzed widespread interest in open-source reasoning models [10–12]. Most existing approaches to improving reasoning rely on supervised fine-tuning (SFT) and rule-based reinforcement learning (RL) [13–16]. For instance, `SkyThought` [17] and `Bespoke` [18] fine-tune models using responses generated by larger reasoning models such as `QwQ-32B` [19] and `DeepSeek-R1`. Other methods like `S1` [20] and `LIMO` [21] highlight the importance of using compact yet high-quality reasoning samples to improve data efficiency. These efforts have led to the creation of numerous open-source reasoning datasets, including `OpenR1-Math-220K` [8], `DeepMath-103K` [4], `OpenThoughts` [5], and others [6, 22, 23]. Notable models trained on these datasets include `OpenR1-Qwen-7B` and `DeepMath-Zero-7B`. Subsequent work [24, 25] has proposed techniques to further optimize SFT and RL pipelines, arguing that structural aspects of reasoning data (e.g., step-wise decomposition) can be more impactful than content alone. More recent studies [3, 26–28] aim to improve the alignment and generalization of reasoning through better data construction, gradient control, and training optimization strategies. Despite these advances, an important factor remains underexplored: the role of `long-context capability` in reasoning. While recent datasets and tasks increasingly involve longer sequences, no existing work systematically investigates how a model's ability to process extended contexts affects its reasoning performance. Our work aims to figure out the relationship between them.

**Long-context ability.** Closed-source models such as `GPT-4` [29], `Claude` [30], and `Gemini` [31] already support context lengths of 128K tokens or more, enabled by advances in both pre-training and post-training [32–35]. In parallel, open-source models—including `LLaMA` [36], `Qwen` [37, 38], and `Phi` [39]—have also extended their capacities, with some reaching 1M tokens, such as `Qwen-2.5-7B-Instruct-1M` [40] and `LLaMA-3.1-Nemotron-8B-UltraLong-4M-Instruct` [41]. To enable long context, several approaches have been proposed. *RoPE-based methods* adapt positional encoding for extrapolation without full retraining, including Position Interpolation [42], NTK Scaling [7], YaRN [43], and SelfExtend [44]. *Attention redesigns* improve scalability or memory retention, such as StreamingLLM [45], LM-Infinite [46], Inf-LLM [47], and Landmark Attention [48]. Another direction is *input compres-*

Table 7: **Validation of the proposed fine-tuning recipe for reasoning.** We evaluate the effectiveness of our proposed training recipe using `Qwen2.5-Math-7B-Instruct`, a model with an initial long-context capacity limited to 4K tokens. We first scale its `RoPE theta` by a factor of 16, and then merge it with `Qwen2.5-7B-Instruct-1M` using a merge ratio of 0.3. We observe consistent improvement in reasoning performance after each step, demonstrating the effectiveness of enhancing long-context capability prior to reasoning fine-tuning to get a higher reasoning ability.

| Operation | Acc(%) | | | | Avg Output Length | | | |
|---|---|---|---|---|---|---|---|---|
| | Base | Short | Long | Avg | Base | Short | Long | Avg |
| MATH500 | | | | | | | | |
| RoPE theta ×1 | **81.88** | 86.28 | 83.80 | 85.04 | 2037 | 1022 | 2147 | 1584 |
| RoPE theta ×16 | 65.16 | 87.68 | 88.72 | 88.20 | 4579 | 4213 | 5789 | 5501 |
| 0.7 × RoPE theta ×16 + 0.3 × 1M-Model | 74.12 | **88.28** | **89.12** | **88.70** | 2228 | **4948** | **6515** | **5731** |
| AIME22-24 | | | | | | | | |
| RoPE theta ×1 | **8.44** | 16.22 | 13.78 | 15.00 | 8257 | 3463 | 10418 | 6940 |
| RoPE theta ×16 | 3.78 | 25.78 | 27.56 | 26.67 | 13411 | 15036 | 17321 | 16179 |
| 0.7 × RoPE theta ×16 + 0.3 × 1M-Model | 7.11 | **26.67** | **29.33** | **28.00** | 6859 | **20265** | **22919** | **21592** |

| (a) MATH500 (Short) | (b) MATH500 (Long) | (c) MATH500 (Avg) |
|---|---|---|

| (d) AIME22-24 (Short) | (e) AIME22-24 (Long) | (f) AIME22-24 (Avg) |
|---|---|---|

Figure 11: Comparison of reasoning accuracy across different configurations. Results are reported on `MATH500` (top row) and `AIME22-24` (bottom row), with separate evaluation for short, long, and average context cases. Our proposed recipe (dark blue bar, dashed line) consistently outperforms the baseline `RoPE theta ×1` and the extended-context baseline `RoPE theta ×16`, demonstrating that enhancing long-context capability prior to reasoning fine-tuning yields stable and significant accuracy improvements, enhancing models' reasoning ability.

*sion*, which reduces sequence length by summarization or filtering [49, 50]. Finally, recent work explores hardware-aligned designs, including sparse attention architectures [51] and MoBA [52].

# 5 Discussion and Conclusions

**Limitations**: Our analysis is limited to 7B–8B models and does not cover larger scales. We also focus on supervised fine-tuning, leaving open how reinforcement learning interacts with long-context ability. Future work should examine these directions across model families and scales.

This paper investigates the overlooked yet crucial role of long-context capability in reasoning performance. Through behavioral analysis and controlled experiments, we demonstrate a consistent correlation between long-context and downstream reasoning ability. Our results show that models with improved long-context capacity not only perform better on tasks involving lengthy inputs but also achieve higher reasoning accuracy even on short-form benchmarks. Furthermore, we find that enhancing long-context ability prior to supervised fine-tuning yields significant gains across multiple reasoning datasets. Based on these findings, we advocate for a training recipe: first, extend the model's long-context ability, then apply reasoning-specific fine-tuning.

## Acknowledgments

This work was supported in part by NSF grants 2112606 and 2117439. Further, this research made use of the High Performance Computing Resource in the Core Facility for Advanced Research Computing at Case Western Reserve University (CWRU). We give special thanks to the CWRU HPC team for their prompt and professional help and maintenance.

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

# A Other Results on Different Long-Context Ability

This section provides additional experimental results that complement the main findings by examining how different levels of long-context ability influence model performance across multiple reasoning benchmarks. While the main paper focused on reasoning accuracy trends, here we provide a broader view including auxiliary benchmarks, output length analysis, and merged-model experiments.

**Effect of RoPE scaling.** We first analyze the impact of varying RoPE $\theta$ scaling factors on `LLaMA3-8B-Instruct`. As shown in Table 8, models with stronger long-context ability generally achieve higher reasoning accuracy across `MATH500`, `AIME22-24`, and `GSM8K`. The trend is most consistent up to RoPE $\times16$, after which performance begins to plateau or slightly decline, suggesting diminishing returns when scaling beyond the effective context length of the training data. We also observe that output length tends to shrink as accuracy improves, consistent with our analysis in Section 3.2, where correct solutions are typically shorter and more concise.

Table 8: **Effect of RoPE $\theta$ Scaling on Long-Context and Reasoning Performance.** Evaluation of `LLaMA3-8B-Instruct` with different RoPE $\theta$ scaling factors on the 32K *Needle-in-a-Haystack* benchmark. `Base` refers to the model's performance before SFT, while `Short` and `Long` denote results after fine-tuning on short and long reasoning datasets, respectively. `Avg` represents the average of the `Short` and `Long` performances. Length represents the average output length.

| RoPE | 32k long | | Acc(%) | | | | Length | | |
|---|---|---|---|---|---|---|---|---|---|
| theta | ctx ability | Base | Short | Long | Avg | Base | Short | Long | Avg |
| | | | | MATH500 | | | | | |
| ×1 | 0 | **24.40** | 50.68 | 58.92 | 54.80 | 746 | 12991 | 11187 | 12089 |
| ×4 | 3.75 | 20.40 | 52.80 | 61.16 | 56.98 | 754 | 9047 | 9949 | 9498 |
| ×8 | 58.30 | 18.96 | 53.12 | 62.24 | 57.68 | 767 | **8476** | 9673 | 9075 |
| ×16 | **77.05** | 17.28 | **54.40** | **64.32** | **59.36** | 608 | 8782 | 9335 | **9059** |
| ×32 | 58.86 | 15.24 | 53.84 | 61.96 | 57.90 | 490 | 8489 | **9526** | 9007 |
| ×64 | 35.00 | 14.20 | 53.28 | 62.36 | 57.82 | 453 | 8605 | 9579 | 9092 |
| | | | | AIME22-24 | | | | | |
| ×1 | 0 | 0.22 | 2.67 | 3.11 | 2.89 | 1895 | 27482 | 26706 | 27094 |
| ×4 | 3.75 | **0.67** | 3.78 | 5.11 | 4.45 | 2048 | 19961 | 20684 | 20322 |
| ×8 | 58.30 | 0.22 | 2.89 | 6.89 | 4.89 | 1723 | **17532** | **17547** | **17539** |
| ×16 | **77.05** | 0.22 | **4.67** | 6.22 | **5.45** | 1617 | 18339 | 18015 | 18177 |
| ×32 | 58.86 | 0.00 | 3.56 | **7.11** | 5.34 | 1372 | 17876 | 18071 | 17973 |
| ×64 | 35.00 | 0.44 | 3.11 | 4.67 | 3.89 | 1108 | 17830 | 18251 | 18041 |
| | | | | GSM8K | | | | | |
| ×1 | 0 | **78.13** | 84.81 | 84.64 | 84.73 | 195 | 2244 | 4769 | 3506 |
| ×4 | 3.75 | 75.54 | 85.64 | 85.88 | 85.76 | 190 | **1973** | 4010 | 2992 |
| ×8 | 58.30 | 73.03 | **85.90** | 85.79 | 85.85 | 184 | 1981 | **3812** | **2897** |
| ×16 | **77.05** | 70.39 | 85.64 | **86.38** | **86.01** | 182 | 2019 | 3936 | 2977 |
| ×32 | 58.86 | 66.35 | 85.41 | 86.14 | 85.78 | 180 | 2026 | 4179 | 3103 |
| ×64 | 35.00 | 62.21 | 85.56 | 85.25 | 85.41 | 182 | 2083 | 4177 | 3130 |

**Extremely long context via model merging.** To test whether extremely long contexts (e.g., 128k or 1M) provide further gains, we construct merged variants of `Qwen2.5-7B-Instruct` by combining its base 32k model with the ultra-long version `Qwen2.5-7B-Instruct-1M`. By adjusting merge ratios, we obtain intermediate models with controllable long-context capabilities, while preserving base reasoning ability. As shown in Figure 9, reasoning performance correlates with effective long-context strength: models with moderate merge ratios (e.g., 0.1, 0.7) achieve consistently strong accuracy, whereas the pure 1M model shows weaker effective long-context utilization and degraded reasoning.

# B  Other Results on Different Merging Ratios

We further analyze how different merging ratios between base and ultra-long variants affect both retrieval ability and downstream reasoning. This provides insight into whether extremely long contexts (e.g., 1M tokens) always yield benefits, or whether moderate ratios strike a better balance between long-context integration and base reasoning stability.

Table 9: **Reasoning Performance After SFT with Different Merge Ratios.** Qwen results are based on merging `Qwen2.5-7B-Instruct-1M` with `Qwen2.5-7B-Instruct`. A ratio of 0.1 means that the long-context (1M) variant contributes 10% to the merged model.

| 1M Merge | 32k long | Acc(%) | | | | Length | | | |
|---|---|---|---|---|---|---|---|---|---|
| Ratio | ctx ability | Base | Short | Long | Avg | Base | Short | Long | Avg |
| MATH500 | | | | | | | | | |
| 0 | 78.1 | **75.00** | 83.48 | 84.84 | 84.16 | 638 | 5572 | 6545 | 6058 |
| 0.1 | 79.1 | 74.64 | 83.56 | **86.28** | **84.92** | 670 | 5625 | 6432 | **6028** |
| 0.7 | 79.5 | 72.80 | **83.88** | 84.88 | 84.38 | 630 | 5812 | 6563 | 6187 |
| 1.0 | 77.7 | 72.16 | 82.28 | 83.48 | 82.88 | 688 | 5897 | 6881 | 6389 |
| AIME22-24 | | | | | | | | | |
| 0 | 78.1 | 8.22 | 17.78 | 23.33 | 20.56 | 1051 | 19642 | 18592 | 19117 |
| 0.1 | 79.1 | **9.33** | 19.11 | 22.89 | 21.00 | 1313 | 19473 | 18178 | **18825** |
| 0.7 | 79.5 | 7.78 | **19.56** | **23.56** | **21.56** | 1416 | 20396 | 18142 | 19269 |
| 1.0 | 77.7 | 7.33 | 16.22 | 20.00 | 18.11 | 1750 | 20507 | 18680 | 19594 |
| GSM8K | | | | | | | | | |
| 0 | 78.1 | **90.75** | **93.19** | 91.54 | 92.37 | 259 | 1429 | 3264 | 2347 |
| 0.1 | 79.1 | **90.75** | 93.15 | 92.16 | 92.66 | 252 | 1472 | 3197 | 2334 |
| 0.7 | 79.5 | 89.1 | 93.07 | **92.27** | **92.67** | 254 | 1488 | 2930 | 2209 |
| 1.0 | 77.7 | 89.13 | 92.84 | 92.34 | 92.59 | 254 | 1494 | 2915 | **2204** |

**Needle-in-a-Haystack (retrieval ability).** We first evaluate the merged models on the 32K *Needle-in-a-Haystack* benchmark. As shown in Figure 12, long-context ability generally increases with higher contribution from the 1M-token variant. However, because most models achieve near-perfect scores, the benchmark does not fully differentiate their effective long-context strength. This motivates evaluating more challenging reasoning datasets where differences manifest more clearly.

**Reasoning benchmarks.** We then fine-tune the merged models on both short and long reasoning datasets and evaluate them on three benchmarks: `MATH500`, `AIME22-24`, and `GSM8K`. Table 9 reports accuracy and output length. Three key observations emerge: (1) Moderate merge ratios (e.g., 0.1 or 0.7) yield the best overall reasoning performance, consistently outperforming both the base (0) and fully merged (1.0) variants. (2) Pure 1M models (ratio 1.0) exhibit degraded reasoning accuracy, despite higher nominal long-context capacity, suggesting weaker effective utilization. (3) Output lengths tend to shrink as reasoning improves, consistent with the observation that successful reasoning requires fewer redundant tokens.

# C  Additional Analysis on Output Length and Context Extension

**Why is the output length decreasing with the long context extension?** The observed decrease in output length is actually aligned with the ratio of correct answers. Correct generations tend to be much shorter and more concise than incorrect ones. With long-context extension, the number of correct answers increases, thus reducing the overall average output length. This phenomenon has also been discussed in prior work [53].

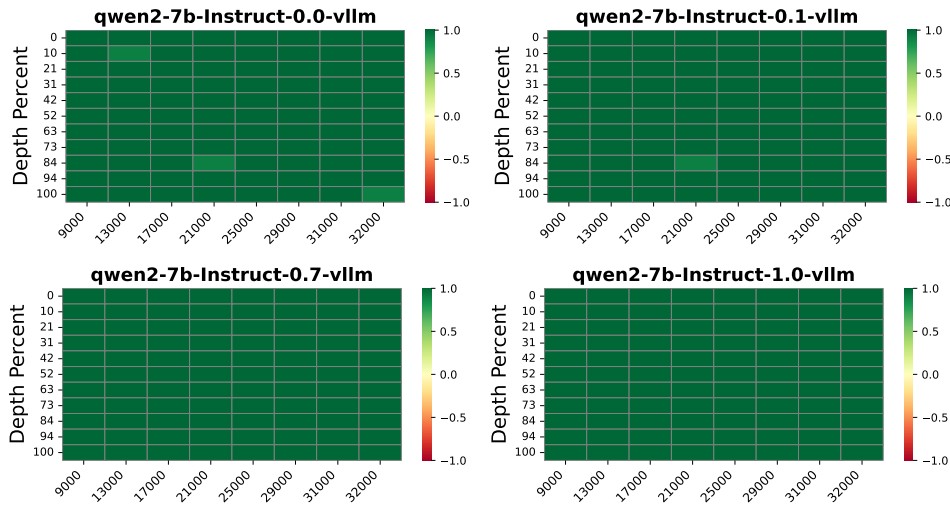

Figure 12: 32K Needle-in-a-Haystack evaluation for different merge ratios. Models with higher long-context contribution achieve stronger retrieval ability, but performance differences saturate quickly.

Table 10: Performance of `Phi-4` under different RoPE scales on AIME.

| Model | RoPE | 32k NIAH (%) | AIME (before, %) | AIME (after, %) |
|-------|------|--------------|------------------|-----------------|
| Phi-4 | ×1   | 52.27        | 14.89            | 47.56           |
| Phi-4 | ×4   | 78.07        | 13.78            | 49.45           |
| Phi-4 | ×16  | 84.77        | 10.22            | 50.17           |

To support this explanation, we report the average lengths of correct and incorrect generations across both short (0–8k) and long (8–16k) training settings for `LLaMA3-8B` under different RoPE configurations: *RoPE* ×1 *(8k context)* and *RoPE* ×16 *(Extended context)*

Table 11: Effect of training length on output length under different RoPE scales.

| RoPE Scale | Setting | Accuracy (%) | Avg Length | # Correct | Avg  | # Wrong | Avg   |
|------------|---------|--------------|------------|-----------|------|---------|-------|
| ×1         | short   | 50.68        | 12991      | 1267      | 3189 | 1233    | 23145 |
| ×1         | long    | 58.92        | 11187      | 1473      | 5928 | 1027    | 20999 |
| ×16        | short   | 54.40        | 8782       | 1360      | 3906 | 1140    | 14921 |
| ×16        | long    | 64.32        | 9335       | 1608      | 6063 | 892     | 17265 |

As shown, models with longer context capacity (e.g., RoPE ×16) achieve higher accuracy while generating shorter outputs on average—primarily because a larger portion of the outputs are correct, and correct answers are generally shorter.

