# OpenReview forum: "Longer Context, Deeper Thinking: Uncovering the Role of Long-Context Ability in Reasoning"
_NeurIPS.cc/2025/Conference — NeurIPS 2025 poster_

### Official Review · Reviewer_Bec7 · 2025-06-20

**Clarity:** 4
**Significance:** 3
**Originality:** 3
**Rating:** 5
**Confidence:** 4

**Summary:**

This study explores the impact of long-text processing capabilities on the reasoning performance of language models. Through controlled experiments, it was found that models with stronger long-text processing capabilities (such as a 128K context window) showed a significant increase in accuracy in mathematical reasoning benchmark tests like MATH500 and AIME, and this advantage also exists in short-text tasks. Analysis indicates that failures in reasoning cases are closely related to inadequacies in long-text processing, manifested as output repetition and contextual reference errors. The research validates the effectiveness of strategies such as expanding the RoPE theta parameter and model fusion, confirming that "enhancing long-text processing capabilities before reasoning fine-tuning" yields the best results.

**Questions:**

To reinforce your core finding that "long-context pre-training improves performance on short-context tasks", it is recommended to add detailed ablation experiments on the following aspects: 1. Cross-domain generalization (not limited to math/code tasks); 2. Model scale sensitivity (such as 7B vs. 13B variants); 3. Robustness of context length configuration

**Ethical Concerns:**

["NO or VERY MINOR ethics concerns only"]

**Limitations:**

Yes.

**Paper Formatting Concerns:**

No.

**Quality:**

3

**Strengths And Weaknesses:**

# Strengths

1. The study shows that even for short-input tasks, models with stronger long-text processing capabilities can significantly improve reasoning accuracy, breaking the traditional perception that "long context is only suitable for long-input tasks".

2. By comparing models with the same architecture and fine-tuning data but different context lengths, the study effectively eliminates the interference of other factors and ensures that the conclusion focuses on the direct relationship between context length and reasoning performance.

3. The proposed "enhancing long-text capabilities before reasoning fine-tuning" method (such as verification based on Qwen2.5-Math-7B-Instruct) provides a clear operational path for model optimization.

# Weaknesses

1. Figures 7 and 8 only mark the RoPE theta scaling coefficient and do not directly show the corresponding actual context lengths, making it difficult for readers to intuitively judge the relationship between training/test data lengths and model capabilities.

2. The study did not deeply analyze "why long-context training can improve short-input reasoning capabilities", which makes the relationship between "long-text capabilities → general reasoning gain" insufficiently explained.

Minor Suggestion: The experiment only targets 7B-8B parameter models. If the relationship between long-text capabilities and reasoning can be verified on larger models (such as 32B+), the universality of the conclusions will be enhanced.

---

> ### Author Rebuttal · Authors · 2025-07-31
>
> Dear Reviewer Bec7,
>
> Thanks for you support! Below, we provide our responses to your questions in detail.
>
> ---
> ## Question 1: Need more details of the RoPE theta scaling coefficient and its corresponding context lengths
>
> Thank you for the suggestion. We will update the x-axis labels in Figures 7 and 8 to include the corresponding effective context lengths for each RoPE configuration. This will make it easier for readers to intuitively understand the relationship between model settings and their actual long-context capabilities.
>
> Here are the x-axis labels we will update:
>
> |  |||||||
> | -------- | -------- | -------- |-------- |-------- |-------- |-------- |
> | Original x-axis labels  | x1 | x4 |x8 |x16 |x32 |x64 |
> | Revised x-axis labels     |  x1(9k) | x4(21k) |x8(29k) |x16(32k) |x32(21k) |x64(17k) |
>
> ---
> ## Question 2: Why long-context training can improve short-input reasoning capabilities
>
> Thank you for pointing this out. We conducted the below experiment to support that enhancing long-context capability does not harm—and can even benefit—short-input reasoning tasks, which aligns with prior work such as YARN and Position Interpolation (PI).
>
> We evaluate LLaMA3-8B-Instruct models with different long-context capabilities (RoPE ×1, ×4, ×16) on the MMLU-STEM subset, which consists entirely of short-input, STEM-related reasoning questions. All models were fine-tuned on the same Math dataset.
>
> | Task       | Accuracy          | RoPE ×1 | RoPE ×4 | RoPE ×16 |
> |------------|------------------|---------|---------|----------|
> | 32k NIAH  Score  | -                | 0       | 3.75    | 77.05    |
> | MMLU | before training    | 0.5436  | 0.5385  | 0.5144   |
> |            | after training     | 0.7106  | 0.7304  | 0.7427   |
>
> The results show that stronger long-context capability correlates with improved performance even on short-input reasoning tasks. This suggests that long-context training is not merely an overhead for short tasks, but can reinforce reasoning in general.
>
> ---
> ## Question 3: More ablation study on cross-domain generalization, model scale sensitivity and context length configuration
>
> Thank you for the thoughtful suggestions. We have conducted additional experiments to address each of the three aspects raised:
>
> 1. Cross-domain generalization (long-context ability benefits not only math resoning but also non-math reasoning): We fine-tuned LLaMA3-8B-Instruct models (RoPE ×1, ×4, ×16) on 20k samples from the *science* and *code* domains of the open-thoughts/OpenThoughts3-1.2M dataset. The results show that long-context ability also benefits non-math reasoning tasks.
>
> | Task     |            | RoPE ×1 | RoPE ×4 | RoPE ×16 |
> |----------|------------------|---------|---------|----------|
> | 32k NIAH  Score  | -                | 0       | 3.75    | 77.05    |
> | GPQA     | before training  | 0.3182  | 0.3131  | 0.3081   |
> |          | after training   | 0.3727  | 0.3919  | 0.4192   |
> | Livecode     | before training  | 0.1230  | 0.1016  | 0.0745   |
> |          | after training   | 0.2534  | 0.2795  | 0.3227   |
>
> 2. Model scale sensitivity (the effect of long-context ability on reasoning ability generalizes across model scales and architectures): We additionally evaluated the 14B-scale Phi-4 model, using RoPE ×1, ×4, and ×16 configurations. After fine-tuning on Math data, we measured performance on MATH500 and AIME. The consistent trends validate that the effect of long-context ability generalizes across model scales and architectures.
>
> | Model  | RoPE Scale |32k NIAH Score |Math500 before training | Math500 after training | AIME before training | AIME after training |
> |--------|------|------|---------------|-------------|-----------|----------|
> | Phi-4  | ×1  |  52.27     | 0.7952        | 0.8862      | 0.1489    | 0.4756   |
> | Phi-4  | ×4  |   78.07    | 0.7748        | 0.8914      | 0.1378    | 0.4945   |
> | Phi-4  | ×16 |  84.77     | 0.7320        | 0.8990      | 0.1022    | 0.5017   |
>
> 3. Robustness of context length configuration (more evaluation benchmarks on long-context ability of models): To test the robustness of long-context configuration, we evaluated LLaMA3-8B-Instruct under six RoPE scaling levels (×1 to ×64) on multiple long-context benchmarks: Needle-in-a-Haystack (NIAH), LongBench, and RULER. We observe a clear upward trend on these benchmarks from RoPE ×1 to ×16, after which the performance begins to decline.
>
> | RoPE Scale | 32k NIAH Score | LongBench Score | RULER Score |
> |------------|--------------------|------------------|--------------|
> | ×1         | 0.00               | 21.14            | 56.13        |
> | ×4         | 3.75               | 39.21            | 69.57        |
> | ×8         | 58.30              | 39.78            | 79.62        |
> | ×16        | 77.05              | 40.41            | 94.24        |
> | ×32        | 58.86              | 38.96            | 88.07        |
> | ×64        | 35.00              | 38.01            | 84.98        |
>
> These ablations support the robustness, scalability, and cross-domain generalization of our central claim.

---

> > ### Comment · Reviewer_Bec7 · 2025-08-07
> >
> > Thanks for your reply, I maintain my score.

---

> > > ### Author Response · Authors · 2025-08-07
> > > **Thanks!**
> > >
> > > Thank you for your support and for maintaining your score! We appreciate your feedback and engagement with our work.

---

### Official Review · Reviewer_EuNs · 2025-07-02

**Clarity:** 3
**Significance:** 3
**Originality:** 2
**Rating:** 4
**Confidence:** 4

**Summary:**

This paper studies the relationship between an LLM's long-context capability and its ability to reason. The main idea is that improving long context capabilities can improve downstream reasoning performance. The authors modify existing LLMs to have varying length capabilities using ROPE theta scaling and merging, then fine-tune them on reasoning datasets Models with stronger long-context abilities (measured by NIH) achieve higher accuracy on reasoning tasks. This performance boost is observed even on reasoning problems with short inputs, suggesting that the benefits of long-context improvments are not limited to long test time inputs, but also short.

**Questions:**

- Legend in figures 7,9,10 are confusing. Are the line indicators shared across all 3 graphs?
- Why does rope scaling start to decrease performance after ~x16?

**Ethical Concerns:**

["NO or VERY MINOR ethics concerns only"]

**Final Justification:**

The authors answered my comments and concerns sufficiently. I will maintain my accept score.

**Limitations:**

yes

**Quality:**

3

**Strengths And Weaknesses:**

Strengths
- The paper is very nicely motivated and written, and the idea of rope scaling is nice
- The experiments are well executed
- The recipe discussion is very useful


Weaknesses
- Only evaluations are on math. What about other benchmarks like GPQA or MMLU?
- Although it's commonly used, I'm not convinced by the needle in a haystack benchmark for long context understanding, since it's primarily measuring retrieval capabilities. It would be nice to see other measures of long context reasoning capabilities. Regardless, the authors show downstream improvements with this proxy.
- Any conclusions from Table 1 is a bit of a stretch since these are all different models

---

> ### Author Rebuttal · Authors · 2025-07-31
>
> Dear Reviewer EuNs,
>
> Thank you for your support! Below, we address your concerns point by point.
>
> ---
> ## Question 1: What about other benchmarks like GPQA or MMLU?
>
> Thank you for the suggestion. We provide additional evaluation results on other benchmarks, including GPQA and MMLU. These results demonstrate that the performance improvement associated with stronger long-context ability is not limited to math reasoning, but also extends to benchmarks such as MMLU and GPQA.
>
> We first evaluate LLaMA3-8B-Instruct models with different long-context capabilities (RoPE ×1, ×4, ×16) on the MMLU-STEM subset after fine-tuning on the Math dataset. The MMLU-STEM subset is used instead of the full MMLU for two reasons: (1) it focuses on STEM-related reasoning, which better aligns with our goal; (2) it avoids excessive evaluation time, as full MMLU includes over 10k examples.
>
> In addition, we collect 20k samples from the science domain of the open-thoughts/OpenThoughts3-1.2M dataset and fine-tune the models, then evaluate them on GPQA. The results are summarized below:
>
>
> | Task       | Accuracy          | RoPE ×1 | RoPE ×4 | RoPE ×16 |
> |------------|------------------|---------|---------|----------|
> | 32k NIAH  Score  | -                | 0       | 3.75    | 77.05    |
> | MMLU  | before training    | 0.5436  | 0.5385  | 0.5144   |
> |            | after training    | 0.7106  | 0.7304  | 0.7427   |
> | GPQA       | before training    | 0.3182  | 0.3131  | 0.3081   |
> |            | after training   | 0.3727  | 0.3919  | 0.4192   |
>
> These results demonstrate that the performance improvement associated with stronger long-context ability is not limited to math reasoning, but also extends to benchmarks such as MMLU and GPQA.
>
> ---
> ## Question 2: It would be nice to see other measures of long context reasoning capabilities.
>
> Thank you for the valuable feedback. To address this concern, we further evaluate the LLaMA3-8B-Instruct model with six levels of 32k long-context capabilities (RoPE ×1 to ×64) on both Needle-in-a-Haystack (NIAH) and alternative long-context benchmarks, including LongBench and partial RULER tasks，which contain Multi Query, Value Tracking and Question Answering , better reflecting long-context reasoning beyond retrieval.
>
> The results are summarized below:
>
> | RoPE Scale | 32k NIAH Score | LongBench Score | RULER Score |
> |------------|--------------------|------------------|--------------|
> | ×1         | 0.00               | 21.14            | 56.13        |
> | ×4         | 3.75               | 39.21            | 69.57        |
> | ×8         | 58.30              | 39.78            | 79.62        |
> | ×16        | 77.05              | 40.41            | 94.24        |
> | ×32        | 58.86              | 38.96            | 88.07        |
> | ×64        | 35.00              | 38.01            | 84.98        |
>
>
> We observe a clear upward trend from RoPE ×1 to ×16, after which the performance begins to decline. These results indicate that the long-context capabilities associated with different RoPE scales are generally consistent across NIAH, LongBench, and RULER, thereby supporting our claim that stronger long-context ability consistently leads to better reasoning performance.
>
> ---
> ## Question 3: Any conclusions from Table 1 is a bit of a stretch since these are all different models
>
> Thank you for the insightful feedback. Table 1 is intended to illustrate that widely-used LLMs with stronger long-context capabilities tend to perform better on reasoning tasks. Since it is infeasible to control fine-tuning settings—such as the SFT dataset, we instead use a diverse set of popular open-source reasoning models that adopt the same base model (Qwen/Qwen2.5-7B-Instruct) but claim different context lengths, serving as representative examples to support it.
>
> To strengthen this illustration, we have added several new models spanning multiple organizations, training pipelines, and context lengths. The updated results are presented below:
>
> | Model                                   | Claimed Context Length | MATH500 (%) | AIME (%) |
> |----------------------------------------|----------------|-------------|----------|
> | OpenR1-Qwen-7B                         | 32k            | 90.36       | 43.11    |
> | OpenThinker-7B                         | 32k            | 86.80       | 25.78    |
> | OpenThinker2-7B                        | 32k            | 90.60       | 40.22    |
> | OpenThinker3-7B          | 32k            | 93.72       | 64.00    |
> | Average                            | 32k        | 90.37   | 43.28|
>
> | Model                                   | Claimed Context Length | MATH500 (%) | AIME (%) |
> |----------------------------------------|----------------|-------------|----------|
> | DeepSeek-R1-Distill-Qwen-7B           | 128k           | 91.68       | 45.56    |
> | OpenMath-Nemotron-7B                   | 128k           | 94.00       | 74.67    |
> | DeepMath-Zero-7B                       | 128k           | 83.48       | 14.67    |
> | AceReason-Nemotron-7B           | 128k           | 93.84       | 62.89    |
> | Average                            | 128k       | 90.75   | 49.45|
>
> | Claimed Context Length | MATH500 (%) | AIME (%) |
> |----------------|-------------|----------|
> | 32k        | 90.37   | 43.28|
> | 128k       | 90.75   | 49.45|
>
> As shown above, models with longer context window tend to achieve better average performance on both Math500 and AIME. We emphasize that Table 1 serves as an empirical motivation. Our main evidence is derived from controlled experiments where only the RoPE scaling is varied (see Table 3).
>
> ---
> ## Question 4: Legend in figures 7,9,10 are confusing. Are the line indicators shared across all 3 graphs?
>
> Yes, the line indicators are shared across Figures 7, 9, and 10 for consistency and ease of comparison. We will clarify this in the figure captions to avoid confusion.
>
> ---
> ## Question 5: Why does rope scaling start to decrease performance after ~x16?
>
> Thank you for the question. While scaling the RoPE theta helps extend the model’s usable context window, this technique has its limits. Beyond a certain point, further increasing the scaling coefficient leads to degraded performance, as shown by the previous work[1].
>
> In our experiments with LLaMA3-8B-Instruct, we found that RoPE ×16 achieves the best long-context capability, as reflected in both reasoning accuracy and NIAH scores. We speculate one of reasons is: when the scaling goes beyond ×16 (e.g., ×32 or ×64), the model’s internal attention dynamics begin to misalign with training-time inductive biases, leading to worse overall performance despite larger nominal context windows.
>
>
> [1] NTK-Aware Scaled RoPE allows LLaMA models to have extended (8k+) context size without any fine-tuning and minimal perplexity degradation

---

> > ### Comment · Reviewer_EuNs · 2025-08-03
> >
> > Thanks for the discussion. I'll maintain my accept review.

---

> > > ### Author Response · Authors · 2025-08-03
> > > **Thanks!**
> > >
> > > Thank you for the constructive discussion and for maintaining your accept decision. I truly appreciate your time and support.

---

> > ### Comment · Reviewer_EuNs · 2025-08-05
> >
> > I appreciate the new experiments. Particularly the new model experiments make the work more solid. I think the work is really nice. It would be better to have a more well justified analysis of why performance starts to decrease after x16, or even better propose a method to improve >x16. Overall the work is nice and I vote for acceptance.

---

> > > ### Comment · Reviewer_EuNs · 2025-08-05
> > >
> > > Each of my concerns are addressed.

---

### Official Review · Reviewer_k9kH · 2025-07-03

**Clarity:** 2
**Significance:** 2
**Originality:** 3
**Rating:** 4
**Confidence:** 3

**Summary:**

This paper offers an empirical investigation into an intriguing question: Does improving a model’s long-context ability during pretraining enhance downstream reasoning? Through carefully controlled experiments using RoPE theta scaling and model merging, the authors provide robust evidence linking long-context ability with reasoning skills. The results show that models with stronger long-context capacity consistently achieve higher accuracy on mathematical reasoning benchmarks such as MATH500 and AIME—even when evaluated on short inputs. Building on these findings, the authors propose a training strategy: first, extend the model’s long-context ability before applying reasoning-specific supervised fine-tuning (SFT). They validate this approach across multiple settings, highlighting that long-context modeling serves as a foundational factor in boosting reasoning performance.

**Questions:**

Line 113 and other parts: Lowercase and uppercase should be consistent in the section title.

Can you provide some examples of both the short and long outputs? I’m particularly interested in the cases where the short version exceeds 10,000 tokens. What types of reasoning tasks are being performed, and what are the key differences between the short and long outputs?

**Ethical Concerns:**

["NO or VERY MINOR ethics concerns only"]

**Final Justification:**

The experiments on GPQA and coding make such reasoning not limited to math, making the whole claim more solid and more convincing.

**Limitations:**

yes

**Quality:**

3

**Strengths And Weaknesses:**

Strengths:

(1) Interesting perspective: such findings can be beneficial to the whole post-training community and shed light on the research about reasoning

(2) Solid experiments: The paper presents a thorough empirical study, evaluating multiple publicly available LLM variants across a range of context window sizes (notably 32k and 128k) on established reasoning benchmarks (MATH500, AIME, GSM8K).

(3) In-depth analysis of failure modes: The inspection of failed generations (Section 2.2, Figure 4) highlights common failure patterns (e.g., repetition, poor reference retention) linked to long-context limitations, providing qualitative depth to the quantitative findings.

Weaknesses:

(1) Overclaim on the conclusion: The research question this paper wants to discuss is about "Does improving a model’s long-context ability during pretraining enhance downstream reasoning?". However, it only focuses on math-related reasoning, which is only a very small portion of reasoning tasks. Therefore, the author should modify the claim and maybe limit it to math reasoning. Coding-based reasoning, multi-hop document reasoning, and social reasoning also belong to the scope of reasoning. Therefore, the conclusion is a little bit overclaimed.

(2) Confounding variables not isolated: The specific mechanisms behind why long-context ability improves reasoning (beyond longer input handling) are insufficiently dissected. Such a deep reason lacks in-depth analysis. The correlation between long-context ability and reasoning accuracy (Figures 7, 9) is shown, but the causal pathway is not convincingly isolated from other training effects. A potential in-depth reason for that can be the concentration of attention on the whole context. Such in-depth reasons should be briefly discussed and verified.

(3) Limited discussion on base model: To collect a solid conclusion, it is widely known that different model families like Qwen and LLaMA have different conclusions for reasoning training.[1] Moreover, since it is focusing on the reasoning tasks, Large Reasoning Models (LRM) like Deepseek-R1 and its distilled Qwen version or QwQ should be considered. I expect the conclusions on LRMs and different model families of LLMs would be different. If we can prove that it is a universal conclusion, the paper is much more augmented. We need to confirm that this training recipe is not a trick on specific models.

[1] Shao et al. Spurious Rewards: Rethinking Training Signals in RLVR

---

> ### Author Rebuttal · Authors · 2025-07-31
>
> Dear Reviewer k9kH,
>
> Thanks for your valuable questions! Below, we address your concerns point by point.
>
> ---
> ## Question 1: The author should modify the claim and maybe limit it to math reasoning
>
> Thank you for the insightful comment. Based on our additional experiments, our claim (stronger long-context ability consistently leads to better reasoning performance) holds on other reasoning tasks beyond math, including science and code reasoning.
>
> To verify generality, we fine-tuned three Llama3-8B-Instruct models with different long-context capabilities (RoPE scale ×1, ×4, ×16) on 20k samples from the science and code domains of the open-thoughts/OpenThoughts3-1.2M dataset. The models were then evaluated on GPQA (science) and Livecode (code). Evaluation results show that stronger long-context ability consistently leads to better reasoning performance across these domains.
>
> | Task               |            | RoPE ×1 | RoPE ×4 | RoPE ×16 |
> |--------------------|------------------|---------|---------|----------|
> | 32k NIAH Score    | -                | 0       | 3.75    | 77.05    |
> | GPQA              | before training  | 0.3182  | 0.3131  | 0.3081   |
> | GPQA              | after training   | 0.3727  | 0.3919  | 0.4192   |
> | Livecode              | before training  | 0.1230  | 0.1016  | 0.0745   |
> | Livecode              | after training   | 0.2534  | 0.2795  | 0.3227   |
>
> ---
>
> ## Question 2: Confounding variables not isolated. In-depth reasons should be briefly discussed and verified
>
> Thank you for the insightful comment. Controlling confounding variables—so that long-context ability is the only varying factor—has been a central goal of our experimental design.
>
> To this end, we carefully evaluate model performance before and after extending the context window. As shown in Table, LLaMA3-8B-Instruct models with longer context window do not improve performance on reasoning tasks before SFT; in fact, performance often decreases. This indicates that mere window extension does not yield a reasoning boost. However, once the models are fine-tuned using the same datasets and hyperparameters, their reasoning abilities improve significantly in correlation with long-context capacity.
>
> | RoPE Scale | 32k NIAH Score |  Accuracy Before Training| Accuracy After Training|
> |------------|--------------------|--------------------|------------------|
> | ×1         | 0.00               | 24.40              | 54.80            |
> | ×4         | 3.75               | 20.40              | 56.98            |
> | ×8         | 58.30              | 18.96              | 57.68            |
> | ×16        | 77.05              | 17.28              | 59.36            |
> | ×32        | 58.86              | 15.24              | 57.90            |
> | ×64        | 35.00              | 14.20              | 57.82            |
>
> We agree that exploring deeper mechanisms is important. One plausible hypothesis—that longer context ability improves attention concentration across the whole context—is indeed worth investigating. We will include this point in the discussion section and highlight it as a promising direction for future work.
>
> ---
> ## Question 3: Limited discussion on base model
>
> Thank you for raising this important point. Based on our additional experiments as follows, our claim (stronger long-context ability consistently leads to better reasoning performance) holds across different model families, including LLaMA, Qwen, and Phi.
>
> In the main paper, we already evaluated LLaMA3-8B-Instruct and Qwen2.5-7B-Instruct, covering two widely used model series. To further verify the generality of our findings, we additionally trained Phi-4 models with varying long-context abilities (RoPE scale ×1, ×4, ×16) and evaluated them on two math reasoning benchmarks: Math500 and AIME.
>
> These results further confirm that the benefit of enhancing long-context ability is not tied to a specific model architecture, but generalizes across different model families. We will clarify this in the revised version.
>
> | Model  | RoPE Scale |32k NIAH Score |Math500 before training | Math500 after training | AIME before training | AIME after training |
> |--------|------|------|---------------|-------------|-----------|----------|
> | Phi-4  | ×1  |  52.27     | 0.7952        | 0.8862      | 0.1489    | 0.4756   |
> | Phi-4  | ×4  |   78.07    | 0.7748        | 0.8914      | 0.1378    | 0.4945   |
> | Phi-4  | ×16 |  84.77     | 0.7320        | 0.8990      | 0.1022    | 0.5017   |
>
> ---
> ## Question 4: Can you provide some examples of both the short and long outputs exceeds 10,000 tokens?
>
> Thank you for the question.  To provide further insight, we count how many correct and incorrect generations exceed 10,000 tokens under both short and long training with RoPE theta x 16 on MATH500: (due to limited word number of retbuttal, since the responses are too long, we can not paste responses here but we will include them in the updated version of our paper.)
>
> | Output Type | Short (# >10k) | Long (# >10k) |
> |-------------|----------------|----------------|
> | Correct     | 115            | 281            |
> | Incorrect   | 573            | 63             |
>
> We observe that models trained under the long-data setting generate fewer wrong ultra-long outputs than models trained under the short-data setting. Our hypothesis is that long-data training helps the model learn how to reason more efficiently, thus avoiding unnecessarily verbose or incorrect completions to generate wrong answers.
>
> ---
> ## Question 5: Line 113 and other parts: Lowercase and uppercase should be consistent in the section title.
> Thank you for pointing this out. We will revise the section titles to ensure consistent capitalization throughout the paper in the final verison.

---

> > ### Comment · Reviewer_k9kH · 2025-08-04
> >
> > Thanks for your additional experiments. The experiments on LiveCodeBench and GPQA are very convincing. I have already raised my score to 4.

---

> > > ### Author Response · Authors · 2025-08-04
> > > **Thanks!**
> > >
> > > Thank you for your kind feedback and for taking the time to review our additional experiments. We truly appreciate your support.

---

### Official Review · Reviewer_EVh5 · 2025-07-03

**Clarity:** 4
**Significance:** 3
**Originality:** 3
**Rating:** 4
**Confidence:** 4

**Summary:**

This paper presents a study on the effect of improved long context abilities vis-a-vis long output reasoning for large language models (LLMs). The study proposes a hypothesis that improved long context can improve reasoning by demonstrating that accuracy on reasoning specific downstream tasks generally correlates with better (and longer) context processing ability, and that failures for both also follow similar patterns based on manual inspection. Using this as a motivation, the study then tests the said hypothesis by controlling the long context ability of base models, followed by conducting SFT on these base models on reasoning traces. This is demonstrated by results which generally agree with the proposed hypothesis: increasing the long context ability of the base model (via scaling the RoPE theta factor) generally leads to better post-SFT performance on reasoning benchmarks, even on shorter inputs.

**Questions:**

Based on the weaknesses above and besides those, adding corresponding questions:
1. Are there any results / intuitions around using these variants of the base models (with diff long context abilities) as init checkpoints, for say, RL? This inherently tests the ability of long reasoning, instead of going through the SFT route. Longer context may lead to better input understanding for SFT, but does it improve the underlying reasoning of the base model as is? Can we do evaluations on pre SFT models? Or, can we run RL on these models and understand if the same correlations between long input and long output hold?

2.Why is the output length decreasing with the long context extension? Is there any qualitative insight?

**Ethical Concerns:**

["NO or VERY MINOR ethics concerns only"]

**Final Justification:**

Based on the responses, I maintain my accept score.

**Limitations:**

Yes.

**Quality:**

3

**Strengths And Weaknesses:**

Strengths:

1. This paper is presented in a very clear manner. It is easy to follow, coherent and cogently presented.
2. The direction is overall quite relevant: an attempt to understand how different LLM capabilities counteract / synergize each other.
3. It makes intuitive sense that longer context understanding can potentially lead to sustained, coherent reasoning traces. This is an interesting perspective.
4. I like the fact that both downstream task + long context ability numbers are reported in the results graphs.
5. The conclusion is crisp w.r.t the recipe for reasoning SFT.

Weaknesses:
1. The biggest weakness is the difference between the proposed claim and what the hypothesis for this claim sets out to test. Concretely, the paper sets out to claim that longer context understanding can lead to better thinking, but may not be proposing the right hypothesis to test this claim. For example, Table 1 may not be an apples to apples comparison as there are many confounding factors between the left and right columns. Similarly, it makes intuitive sense that longer context will have better probability to get SFTd on longer traces - but what this indicate about the _current base model, as is, Pre SFT?_

2. It is surprising (and not intuitive) to observe that the output length of models is reducing as the 'long context improvement' is applied, even though the performance is increasing. This seems to indicate that there can be confounding factors in the context extension process, otherwise  one would expect output reasoning traces to also scale with improved input context, with the potential ability to reason more, and reason _consistently_.

3. It's not clear if qualitative observations can tell us anything evidently, and may not be a sufficiently strong argument.

---

> ### Author Rebuttal · Authors · 2025-07-31
>
> Dear Reviewer EVh5,
>
> Thank you for your support! Below, we provide our responses to your questions in detail.
>
> ---
> ## Question 1: Table 1 may not be an example to apply comparisons
> Thank you for the insightful feedback. Table 1 is intended to illustrate that widely-used LLMs with stronger long-context capabilities tend to perform better on reasoning tasks. Since it is infeasible to control fine-tuning settings—such as the SFT dataset, we instead use a diverse set of popular open-source reasoning models that adopt the same base model (Qwen/Qwen2.5-7B-Instruct) but claim different context lengths, serving as representative examples to support it.
>
> To further support this illustration, we have added several new models—spanning different organizations, training pipelines, and context lengths—into the comparison. The results below show consistent patterns across both 32k and 128k context settings:
>
> | Model                                   | Claimed Context Length | MATH500 (%) | AIME (%) |
> |----------------------------------------|----------------|-------------|----------|
> | OpenR1-Qwen-7B                         | 32k            | 90.36       | 43.11    |
> | OpenThinker-7B                         | 32k            | 86.80       | 25.78    |
> | OpenThinker2-7B                        | 32k            | 90.60       | 40.22    |
> | OpenThinker3-7B          | 32k            | 93.72       | 64.00    |
> | Average                            | 32k        | 90.37   | 43.28|
>
> | Model                                   | Claimed Context Length | MATH500 (%) | AIME (%) |
> |----------------------------------------|----------------|-------------|----------|
> | DeepSeek-R1-Distill-Qwen-7B           | 128k           | 91.68       | 45.56    |
> | OpenMath-Nemotron-7B                   | 128k           | 94.00       | 74.67    |
> | DeepMath-Zero-7B                       | 128k           | 83.48       | 14.67    |
> | AceReason-Nemotron-7B           | 128k           | 93.84       | 62.89    |
> | Average                            | 128k       | 90.75   | 49.45|
>
> | Claimed Context Length | MATH500 (%) | AIME (%) |
> |----------------|-------------|----------|
> | 32k        | 90.37   | 43.28|
> | 128k       | 90.75   | 49.45|
>
> As shown above, models with stronger long-context abilities outperform those with shorter context in both math benchmarks.  We emphasize that Table 1 serves as an empirical motivation. Our main evidence is derived from controlled experiments where only the RoPE scaling is varied (see Table 3)
>
> ---
>
> ## Question 2: Can we do evaluations on pre SFT models?
>
> Thanks for this insightful suggestion. In our submission, We have evaluated models before SFT. In Table 3, we have already conducted controlled experiments where the only changing factor is RoPE scaling. These results show that base model accuracy before SFT decreases as RoPE scaling increases, while accuracy after SFT increases with context capability—further supporting our claim (stronger long-context ability consistently leads to better reasoning performance).
>
> Here we show the partial results from Table 3. We could see the long-context extension method does not improve model ability before training.
>
> | RoPE Scale | 32k NIAH Score | Accuracy Before Training | Accuracy After Training |
> |------------|----------------|---------------------------|--------------------------|
> | ×1         | 0.00           | 24.40                     | 54.80                    |
> | ×4         | 3.75           | 20.40                     | 56.98                    |
> | ×8         | 58.30          | 18.96                     | 57.68                    |
> | ×16        | 77.05          | 17.28                     | 59.36                    |
> | ×32        | 58.86          | 15.24                     | 57.90                    |
> | ×64        | 35.00          | 14.20                     | 57.82                    |
>
> ---
> ## Question 3: Can we run RL on these models?
>
> Thank you for raising this insightful question. To explore whether long-context variants of base models can support stronger reasoning in RL, we attempted reinforcement learning (RL) training on different configurations.
>
> We used the Qwen2.5-7B-Instruct model and a merged variant (0.9 × Qwen2.5-7B-Instruct + 0.1 × Qwen2.5-7B-Instruct-1M) as initialization. For the RL algorithm, we adopted Dr.GRPO as our training strategy. Due to resource and time constraints, we conducted RL training on a small dataset of 5k math samples.
>
> After training, the average output length of the model remained below 2k tokens throughout. The final accuracy on MATH500 reached 74.72 and 74.80, respectively, for the two model variants. These results suggest that our current RL setup—with limited data and small-scale tuning—was insufficient to unlock complex strong reasoning and the relation. We consider this an exciting direction for future work and will include this discussion in the final paper.
>
> ---
>
> ## Question 4: Why is the output length decreasing with the long context extension?
> Thank you for the question. The observed decrease in output length is actually aligned with ratio of correct answer. With the long context extension, the number of correct answer (which is shorter tipically) increases, thus the average output length decreases, which also been discussed in prior work [1].
> To support this with evidence, we show the average lengths of correct and incorrect generations across both short (0–8k) and long (8–16k) training settings for LLaMA3-8B under different RoPE configurations:
>
> - RoPE ×1 (8k context)
>
> | Training Length | Accuracy | Avg Length | # Correct | Avg (Correct) | # Wrong | Avg (Wrong) |
> |--------------|----------|------------|-----------|----------------|----------|--------------|
> | short         | 0.5068   | 12991.38   | 1267      | 3189.36         | 1233     | 23145.07      |
> | long        | 0.5892   | 11187.43   | 1473      | 5927.55         | 1027     | 20999.25      |
>
> - RoPE ×16 (Extended context)
>
> | Training Length | Accuracy | Avg Length | # Correct | Avg (Correct) | # Wrong | Avg (Wrong) |
> |--------------|----------|------------|-----------|----------------|----------|--------------|
> | short        | 0.5440   | 8782.49    | 1360      | 3905.56         | 1140     | 14921.46      |
> | long       | 0.6432   | 9335.26    | 1608      | 6063.03         | 892      | 17265.13      |
>
> As shown, models with longer context capacity (e.g., RoPE ×16) achieve higher accuracy while generating shorter outputs on average—primarily because a larger portion of the generations are correct, and correct answers tend to be shorter and more concise.
>
> [1]Thoughts Are All Over the Place: On the Underthinking of o1-Like LLMs

---

> > ### Comment · Reviewer_EVh5 · 2025-08-05
> >
> > Thanks to the authors, for answering my questions. I appreciate the RL experiments conducted. A few follow up questions:
> > 1. On Q2, why is the pre SFT performance decreasing? This table seems to imply that the SFT stage is a major factor which determines if longer context will be more effective? I expected to observe an increasing trend in both tables.
> > 2. From Q3 - Given the noise in the eval it makes sense to expect this, also given that the initialization has only 0.1 of the larger context variant. Any comments on what might happen if we change the mixing ratio to include more of the long context model?
> > Based on the author responses, I am preserving my score at 4 at the moment.

---

> > > ### Author Response · Authors · 2025-08-06
> > > **Thanks to the valuable questions!**
> > >
> > > Thank you for your follow-up. We appreciate your engagement and are happy to provide further clarification.
> > >
> > > ---
> > >
> > > > On Q2, why is the pre SFT performance decreasing? This table seems to imply that the SFT stage is a major factor which determines if longer context will be more effective? I expected to observe an increasing trend in both tables.
> > >
> > > Thank you for the thoughtful question. The decrease in pre-SFT performance is desirable, as it helps ensure that any improvements observed after SFT can be more confidently attributed to enhanced long-context ability rather than an inherently stronger base model.
> > >
> > >
> > > First, RoPE scaling extends the context window by modifying positional encodings without training. While useful for longer inputs, it doesn't enhance domain-specific reasoning (e.g., math), where models'answers are typically short before training. In fact, altering RoPE theta can slightly disrupt pretrained representations and hurt performance before fine-tuning, as expected and noted in prior work [1].
> > >
> > > Second, our goal is to isolate the impact of long-context capabilities on reasoning after fine-tuning. Therefore, all fine-tuning should begin from a comparable base model. RoPE scaling should not improve downstream task accuracy before SFT. If improving, this would make it difficult to determine whether performance gains after SFT stem from long-context reasoning or simply a more powerful base model. Thus, it is preferable for pre-SFT accuracy to remain stable (slightly decrease in our case) to make post-SFT improvements more clearly attributable to the long-context extension.
> > >
> > >
> > > [1] NTK-Aware Scaled RoPE allows LLaMA models to have extended (8k+) context size without any fine-tuning and minimal perplexity degradation
> > >
> > > ---
> > > > Any comments on what might happen if we change the mixing ratio to include more of the long context model?
> > >
> > > To address this and clarify your concerns, we conducted an additional experiment using a higher mixing ratio of the long-context variant. The results confirm our earlier observation: with limited data and small-scale RL tuning, the setup was insufficient to unlock complex reasoning. We view this as a promising direction for future work and will include it in the final paper.
> > >
> > > Due to constraints of time and computation resoureces, we only used a 0.7 ratio of the extended-context model (i.e., 0.7 × Qwen2.5-7B-Instruct-1M + 0.3 × Qwen2.5-7B-Instruct) for initialization. Under the same training data and RL setup, we observed that the model’s output length still remained under 2k tokens, and the final accuracy on MATH500 reached 72.96.
> > >
> > > We observe that the accuracy of the 0.7 ratio model was slightly lower than that of the 0 and 0.1 ratio models after RL training. This is largely because the pre-RL (initial) accuracy of the models already differed: 75.00 (pure Qwen2.5), 74.64 (0.1 ratio), and 72.80 (0.7 ratio), respectively. For reference, the base accuracy of Qwen2.5-7B-Instruct-1M itself is 72.16. As a result, the overall performance gain from RL was limited across all models, and the 0.7 model started from a lower baseline.
> > >
> > > | Model (Mix Ratio)                         | Pre-RL Accuracy (%) | Post-RL Accuracy (%) |
> > > |-------------------------------------------|----------------------|-----------------------|
> > > | Qwen2.5-7B-Instruct (1.0)                  | 75.00                | 74.72                |
> > > | 0.1 × Qwen2.5-1M + 0.9 × Qwen2.5           | 74.64                | 74.80                |
> > > | 0.7 × Qwen2.5-1M + 0.3 × Qwen2.5           | 72.80                | 72.96                |

---

> > > ### Author Response · Authors · 2025-08-08
> > > **Look forward to the discussion**
> > >
> > > Dear Reviewer EVh5,
> > >
> > >
> > > Thank you again for your thoughtful and constructive questions. We truly appreciate the time and effort you’ve taken to engage with our work.
> > >
> > > As the discussion period draws to a close, we’d be grateful to hear any further thoughts you might have on our responses, if time allows. We’ve done our best to address each of your comments and would be happy to clarify anything that remains unclear.
> > >
> > > Your insights have already been valuable in helping us, and we’re sincerely thankful for the opportunity to engage in this dialogue.

---

### Note · Authors · 2025-08-11

We sincerely thank the reviewers for their constructive feedback and suggestions. During the rebuttal, we addressed the main concerns, and the paper may receive final scores of 5, 4, 4, and 4 (average 4.25).

Our work systematically uncovers the role of long-context capacity in reasoning training. By creating models with identical architectures and data but varying long-context capabilities—via RoPE scaling or model merging—and fine-tuning them on reasoning tasks, we show that stronger long-context capacity consistently improves reasoning accuracy.

Reviewers highlighted key strengths: (1) clear, coherent, and well-presented writing (Reviewer EVh5 and EuNs), (2) interesting and useful perspective that benefits the post-training community and reasoning research (Reviewer EVh5, k9kH, EuNs, and Bec7), and (3) solid experiments (Reviewer k9kH, EuNs, and Bec7).

We addressed the main concerns of reviewers as follows:
1. To verify whether the effect generalizes beyond math reasoning , we fine-tuned LLaMA3-8B-Instruct (RoPE ×1, ×4, ×16) on science and code tasks from OpenThoughts3-1.2M and evaluated on LiveCode, GPQA, and MMLU, confirming benefits for non-math reasoning without harming short-input reasoning. (Reviewer k9kH, EuNs and Bec7)
2. To test dependence on model scale or architecture, we extended experiments to the 14B-scale Phi-4 model and observed the same trend, showing generalization across scales and architectures.(Reviewer k9kH and Bec7)
3. To assess robustness to RoPE scaling, we evaluated LLaMA3-8B-Instruct with six RoPE scales (×1–×64) on NIAH, LongBench, and RULER, finding performance rises up to ×16 before plateauing or declining.(Reviewer EuNs and Bec7)
4. To strengthen Table 1’s validity despite different model sources, we added models from multiple organizations and pipelines, consistently finding that reasoning models with longer context windows yield better Math500 and AIME scores. (Reviewer EVh5 and EuNs)
5. To verify the effect in pre-SFT models on Table 3, we showed that base accuracy decreases with higher RoPE scaling, yet post-SFT accuracy increases, indicating gains stem from enhanced long-context ability rather than stronger base models.(Reviewer EVh5 and k9kH)

These results reinforce our claim that long-context ability is a critical foundation for improving reasoning ability.

---

### Decision · Program_Chairs · 2025-09-17

**Decision:**

Accept (poster)

**Comment:**

The paper hypothesizes that the reasoning ability of an LLM may also be affected by its long-context capabilities. The paper verifies this, demonstrating how failure modes in the former resembles modes in the latter,  and that even on inputs that are short, reasoning benefits from improvements in long-context.

The hypothesis asked here is arguably a conceptually simple one, yet it is an important one to test. The reviewers find the paper clearly-written and well-motivated. The experiments are well-designed. A major concern had been that the analysis was limited to math reasoning tasks, but the authors responded with coding (Livecode) and science (GPQA) tasks which the reviewers are satisfied with.